# Potential global distribution of *Setaria italica*, an important species for dryland agriculture in the context of climate change

**Jingtian Yang[1], Xue Jiang[2], Yunlong Ma[2], Mei Liu[1], Zixi Shama[1], Jiayi Li[1], Yi Huang[2,3]***

**1** Ecological Security and Protection Key Laboratory of Sichuan Province, Mianyang Normal University, Mianyang, 621000, China, **2** Engineering Research Center for Forest and Grassland Disaster Prevention and Reduction, Mianyang Normal University, Mianyang, 621000, China, **3** China College of Science, Tibet University, Lhasa, 850012, China

* hyhy1232021@163.com

**Data Availability Statement:** All relevant data are within the manuscript and its Supporting Information files.

## Abstract

*Setaria italica* (*S. italica*, Linnaeus, 1753) is a drought-resistant, barren-tolerant, and widely adapted C-4 crop that plays a vital role in maintaining agricultural and economic stability in arid and barren regions of the world. However, the potential habitat of *S. italica* under current and future climate scenarios remains to be explored. Predicting the potential global geographic distribution of *S. italica* and clarifying its ecological requirements can help promote sustainable agriculture, which is crucial for addressing the global food crisis. In this study, we predicted the potential global geographic distribution of *S. italica* based on 3,154 global distribution records using the Maxent model and ArcGIS software. We assessed the constraints on its potential distribution based on the contribution of environmental factors variables. The predictive accuracy of the Maxent model was evaluated using AUC values, TSS values, and Kappa statistics, respectively. The results showed that the Maxent model had a high prediction accuracy, and the simulation results were also reliable; the total suitable habitats of *S. italica* is $5.54×10^7$ km$^2$, which mainly included the United States (North America), Brazil (South America), Australia (Oceania), China, India (Asia), and the Russian Federation (Europe). The most suitable habitat of *S. italica* was $0.52×10^7$ km$^2$, accounting for 9.44% of the total areas, mainly in the United States, India, the Russian Federation, and China. Soil and precipitation (driest monthly precipitation, hottest seasonal precipitation) are the most critical factors limiting the potential distribution of *S. italica*. Compared with the modern potential distribution, we predict that the four future climate change scenarios will result in varying reductions in the possible geographic ranges of *S. italica*. Overall, climate change may significantly affect the global distribution of *S. italica*, altering its worldwide production and trade patterns.

## Introduction

Discussing and studying the factors affecting crop growth, development, and distribution has become a hot scientific issue in the past decades [1–4]. Crops are deeply connected to social

**Funding:** This work was supported by grants from The Scientific research initiation project of Mianyang Normal University (QD2019A13, QD2021A37 and QD2023A01; URL: https://www.mtc.edu.cn/), the Funding of the Open Project from the Ecological Security and Protection Key Laboratory of Sichuan Province (ESP1608, ESP2008, ESP2201 and ESP2204; URL: https://zdsys.mtc.edu.cn/), the Sichuan Provincial Education Department Scientific Research Project (15ZB0283; URL:http://edu.sc.gov.cn/), the Sichuan Provincial Science and Technology Department Project (2023NSFSC0750; URL: https://kjt.sc.gov.cn/). The funders had a role in study design, data collection and analysis, decision to publish, or preparation of the manuscript. There was no additional external funding received for this study.

**Competing interests:** The authors have declared that no competing interests exist.

and agricultural advancements as a unique plant resource [2,5]. Climate, topography, hydrology, and soil composition influence crop growth and development [1,6]. Of these factors, non-climatic factors dominate short-term biological changes, while climate change is the key factor that significantly affects growth, development, and optimal distribution [6–8]. There is a scientific consensus that climate change is a crucial driver of agricultural progress [1,4,9]. Studies have shown that global warming severely affects crop yields and threatens global food security and sustainable development [10]. Basic food self-sufficiency must be firmly achieved as a strategic measure to maintain international peace and security [11]. Therefore, in the context of global warming, a systematic exploration of the geographical distribution patterns of crops and their response to climate change becomes essential. This research provides a scientific basis for production layout and cultivation management and establishes a basic framework for regional sustainable development, ecological management, and environmental governance. In the face of changing climatic conditions, the significance of this research will significantly extend to ensuring global food security.

Species Distribution Models (SDMs) have been widely used to study the effect of climate change on the potential geographic distribution of species. These models play an increasingly critical role in predicting species distribution by considering changes in various environmental factors [6,12]. Over ten distinct models, including Bioclim (Bioclimatic Envelope Models), Domain (Domain Models), GARP (Genetic Algorithm for Rule-set Production), MaxEnt (Maximum Entropy Model), etc., have been applied in different fields such as plant protection, utilization, and pest invasion [12–18]. Notably, the MaxEnt model has several advantages over its counterparts: (1) it requires only a single distribution record during modeling, (2) it accommodates both continuous and discrete variables, (3) it is simple to operate and does not require complex format conversions of species distribution and environmental data, (4) a minimal sample of species "presence" data yields highly effective simulations, and (5) its theoretical foundations are closely related to ecological principles, enhancing understanding of species suitability [1,3,19,20]. Due to the differences in theoretical foundations among various models, their simulation and prediction performances exhibit considerable variation. Among these models, MaxEnt has consistently demonstrated superior simulation results [8,21]. Therefore, the MaxEnt model is particularly suitable for assessing the potential global distribution of *S. italica* because its simulation results are reliable and ecologically compatible.

The species *S. italica* is an annual herbaceous plant in the family Gramineae with similar characteristics to millet (Fig 1). It was one of the first domesticated crops globally and is highly resilient to drought, barrenness, and high temperatures [22–26]. *S. italica* is nutritious and rich in natural active substances with hypoglycaemic, hypolipidemic, and antioxidant properties and is also a source of C4 biofuel, which can be processed into a myriad of food and industrial products (Fig 1D) [23,27]. *S. italica* has gained popularity due to its high yield, making it the most crucial miscellaneous crop globally [23,28]. Since much of the global population has a high-protein diet, including *S. italica* in recipes can help promote balanced nutritional intake [23]. In essence, the significance of *S. italica* goes beyond its economic contribution and has important implications for maintaining human well-being and global food security. Despite its agricultural prominence, a noteworthy gap exists in potential distribution studies for *S. italica*. Studies have shown that ecological modeling studies that consider environmental variables provide practical insights for predicting potentially suitable distributions of crops. Still, similar studies for *S. italica* need to be more robust [1,4,5,29]. Such research is expected to contribute meaningfully to solving the current global food crisis by advancing agriculture and addressing critical human life and health issues.

Due to the lack of specific studies on the potential geographic distribution of *S. italica*, using the MaxEnt model is an appropriate approach. While many studies have applied various

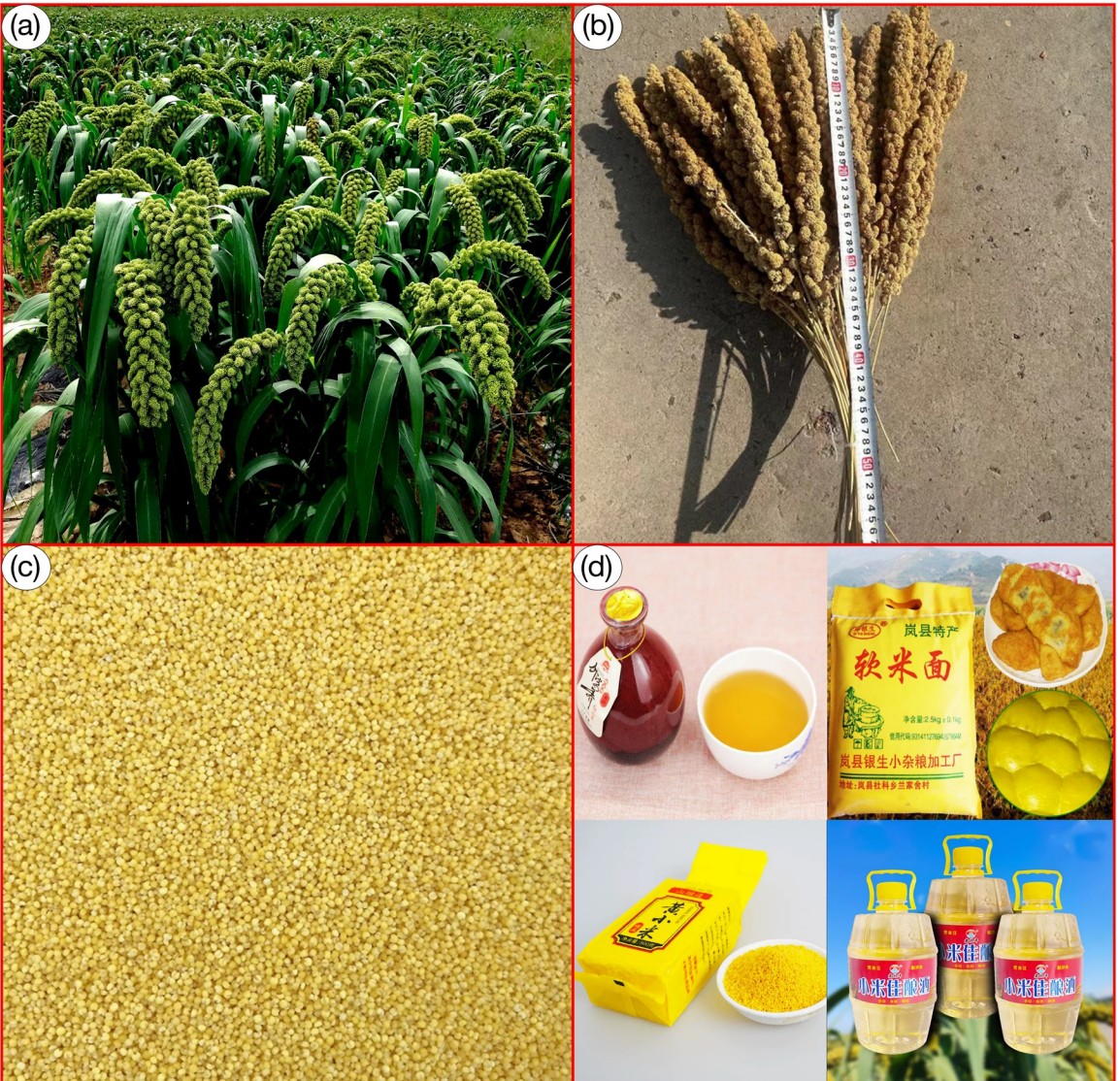

**Fig 1. Stages of S. italica processing.** (a) Unprocessed, (b) Harvesting, (c) After shelling, (d) Processed products.

models to predict crop distributions, specific considerations for *S. italica* have been notably lacking. The cultivation of *S. italica* currently confronts practical challenges, including the unknown potential planting areas and the imperative to comprehend its distribution dynamics in response to climate change within the context of global warming.

This study aims to address these challenges by predicting the potential distribution of *S. italica* across different periods, as illustrated in Fig 2; the objectives of this study were to (1) reveal the present and future distribution patterns of *S. italica*, delineate trends in its potential distribution under the influence of global warming; (2) Demonstrated the changing trends of potential distribution of *S. italica* under the background of climate change; (3) Explored the responsive relationship of *S. italica* distribution to climate change. By delving into these aspects, the research aims to contribute valuable insights for the theoretical underpinning of *S. italica* cultivation, facilitating informed decision-making in the face of uncertainties associated with global warming.

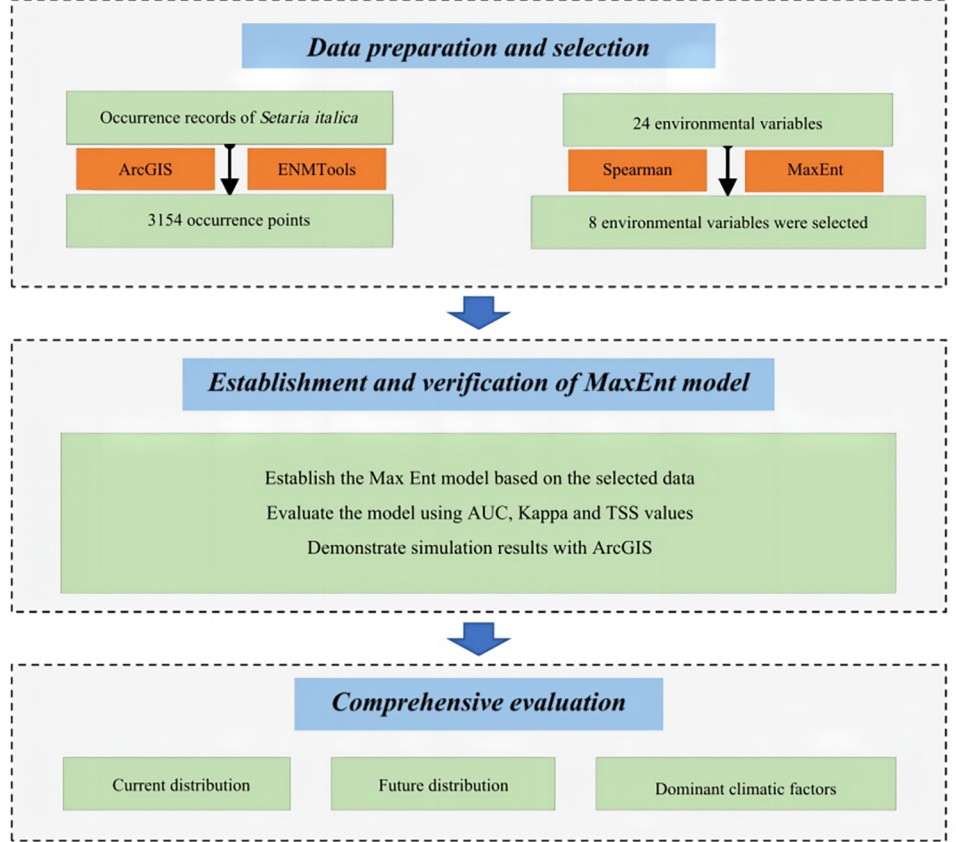

**Fig 2. Flowchart displaying the steps of the present study.**

## Materials and methods

### Species data sources

To obtain the occurrences of *S. italica*, we conducted field surveys. We made queries in databases such as the Global Biodiversity Database (http://www.gbif.org/, last accessed on April 2023), China Plantwise (http://www.iplant.cn, last accessed on April 2023), and China Digital Herbarium (https://www.cvh.ac.cn, last accessed on April 2023) [15,18], 8566 occurrence records were collected (S1 Table). In this paper, the Coordinates for *S. italica* were confirmed and filtered to exclude records with insufficient or redundant descriptions. To avoid the uncertainty of overfitting the model due to the concentration of distribution data in certain areas, the buffer zone tool in ArcGIS 10.2 was used in this study. Following the resolution of environmental variables, a buffer zone with a radius of 5 km was established around each distribution point, allowing for only one distribution point within 5 km. In the end, 3154 valid distribution points of *S. italica* were obtained, and a distribution map (Fig 3) was generated. All records were imported into Microsoft Excel 2020 and saved in "CSV" format.

### Sources of environmental variables

A comprehensive analysis of the potential distribution of *S. italica* was conducted in this study. Future climate data for the 2050s (2041–2060) and 2070s (2061–2080) under four representative concentration pathways (SSP1-2.6, SSP2-4.5, SSP3-7.0, and SSP5-8.5) were generated

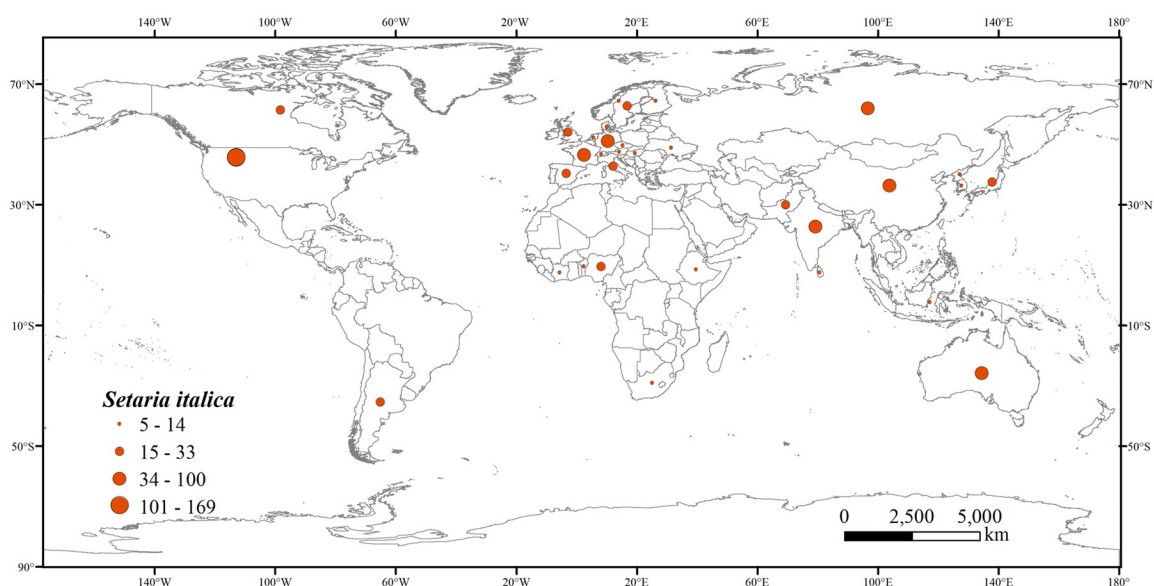

**Fig 3. Global occurrence records of *S. italica*.** World administrative maps were acquired from the free National Earth System Science Data Center (http://www.geodata.cn).

using the Community Climate System Model (GCSM) 4.0. Climate and digital elevation data for current and future emission scenarios were retrieved from the WorldClim database (DEM, http://www.worldclim.org//, last accessed on April 2023). Soil data corresponds to the Coordinated World Soil Database (HWSD V1.2, https://www.fao.org/, last accessed on April 2023), ensuring a robust understanding of the soil characteristics. UV-B radiation data, a crucial environmental factor, was sourced from the global UV-B radiation dataset (gIUV, https://www.ufz.de/gluv/index.php, last accessed April 2023).

To enhance the spatial context of our study, we incorporated world administrative maps obtained from the National Earth System Science Data Center (http://www.geodata.cn). The integration of these datasets was achieved through ArcGIS software (version 10.2), facilitating a unified representation at a five arc-minute resolution [18].

Considering the potential interdependence of environmental variables, we conducted a Spearman correlation analysis (S2 Table), as depicted in Fig 4. This analysis allowed us to discern correlations among the variables before incorporating them into the MaxEnt model. When the correlation coefficient between two environmental factors exceeded or equaled 0.8, the one with a higher contribution rate was retained. As a result of this selection process, eight key environmental variables were identified and utilized in subsequent MaxEnt model runs, ensuring the model's robustness and effectiveness in predicting the potential distribution of *S. italica* [8,18].

## Model construction and evaluation

The MaxEnt (MaxEntV3.4.1, http://biodiversityinformatics.amnh.org/open.source/MaxEnt/) modeling process involved importing distribution point and environmental factor data for *S. italica*, which were divided into a test set (25%) and a training set (75%) to ensure the accuracy of our predictions. To address the inherent challenge of false positives and negatives in species distribution models, rigorous evaluation metrics were employed to assess the usability and accuracy of the model [3,8].

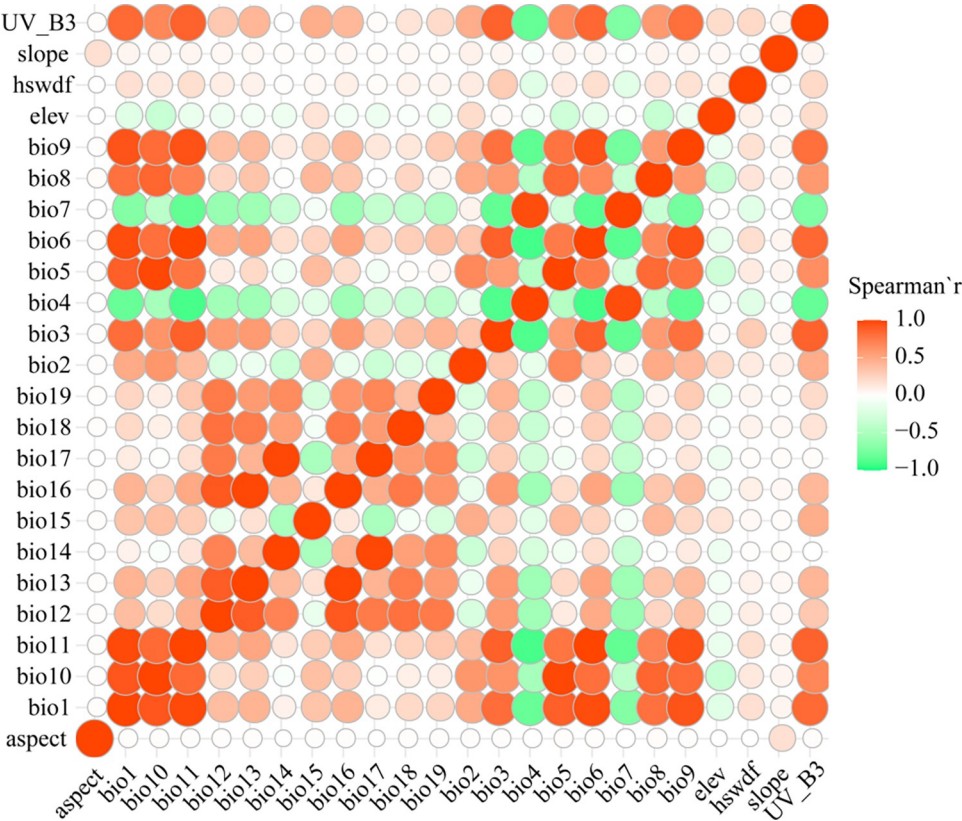

**Fig 4. Heat map for correlation analysis of environmental variables.**

Commonly utilized theoretical evaluation metrics were considered, including overall accuracy, sensitivity, specificity, AUC values, TSS values, and Kappa statistics. For enhanced credibility, this study selected AUC values, TSS values, and Kappa statistics for model accuracy evaluation across three assessment methods. The accuracy of the evaluated model was categorized into five levels: failure, poor, average, good, and very good. Specifically, intervals for AUC values were defined as follows: (AUC $\leq$ 0.60, failure), (0.60 < AUC $\leq$ 0.70, poor), (0.70 < AUC $\leq$ 0.80, average), (0.80 < AUC $\leq$ 0.90, good), and (0.90 < AUC $\leq$ 1.00, very good). Similarly, Kappa values were categorized into intervals as follows: (0.40 < Kappa $\leq$ 0.55, poor), (0.55 < Kappa $\leq$ 0.70, fair), (0.70 < Kappa $\leq$ 0.85, good), and (0.85 < Kappa $\leq$ 1.00, very good) [30–32].

This study used experimental models with different parameter settings to assess performance. These models were also optimised based on observed responses to known Italian bat distribution sites and their corresponding environmental factors. The adjustment process involved establishing the regularization multipliers (RM) range from 0.5 to 4. Additionally, six feature combinations (FC) were employed to optimise the model parameters: L (linear features), LQ (linear features + quadratic features), H (hinge features), LQH (linear features + quadratic features + hinge features), LQHP (linear features + quadratic features + hinge features + product features), and LQHPT (linear features + quadratic features + hinge features + product features + threshold features). After careful evaluation, the optimal combination for the regularisation multiplier and feature selection was RM and LQHPT, respectively [32].

## Selection of dominant environmental variables

Determining dominant factors in environmental studies is marked by varying perspectives among researchers. Presently, a prevalent approach involves establishing dominance based on the contribution rate of environmental factors. This entails considering factors as dominant once they surpass a certain contribution rate threshold. However, the diversity in criteria arises due to the subjective nature of threshold selection. In this study, we adopted a specific criterion for identifying dominant environmental factors. The top three environmental factors with the highest contribution rates were selected as dominant factors, as illustrated [32].

## Classification of potentially suitable areas

The MaxEnt model produced an ASCII raster layer depicting the probability (P) of *S. italica* presence, ranging from 0 to 1. To categorize these probabilities into habitat suitability classes, we followed the IPCC-CMIP6 guidelines [6,30]: A probability greater than 0.5 corresponds to a most suitable habitat; The range of $0.3 \leq P < 0.5$ designates a moderately suitable habitat; For probabilities within $0.1 \leq P < 0.3$, the habitat is considered low suitable.

## Drawing the potential geographical distribution

As per previous literature [33,34], suitable areas were coded as one and the unsuitable regions as two under current conditions. Under future conditions, suitable areas were coded as three and the unsuitable regions as 4. The current and future conditions data were multiplied in ArcGIS 10.2, where a cell value of 3 indicated an unsuitable area and 8 signified a suitable area. Additionally, 4 showed an increased suitability area, while 6 signified a decreased suitability area under future conditions. Finally, habitat areas were computed after coordinate projection.

# Results

## Model accuracy evaluation

Based on the results (Fig 5), we can conclude that the MaxEnt model predicted a "very good" AUC value, and the Kappa and the TSS statistics indicated "good" performance; these evaluations confirm the model's appropriate configuration and high reliability for subsequent analysis.

## Current potential geographical distributions of *S. italica*

The potentially suitable habitats for *S. italica* covered an area of $5.54 \times 10^7$ km$^2$ (Fig 6) and are mainly distributed in the United States (North America), Brazil (South America), Australia (Oceania), China, India (Asia), and the Russian Federation (Europe) (Fig 7). Within this range, highly suitable habitats occupied $0.52 \times 10^7$ km$^2$, constituting 9.44% of the total area (Fig 6). Notably, these highly suitable habitats were concentrated in the eastern United States, the southern part of India, and the western part of the Russian Federation (Fig 7). Moderately suitable habitats cover an area of $1.31 \times 10^7$ km$^2$, accounting for 23.73% of the total suitable area (Fig 6). These habitats were primarily observed in the Russian Federation, the United States, China, and India (Fig 7). The area of low suitable habitat is $3.70 \times 10^7$ km$^2$, accounting for 66.83% of the total suitable area (Fig 6). These areas were predominantly found in China, the United States, the Russian Federation, Brazil, Australia, and other countries (Fig 7). In summary, the potential geographic range of *S. italica*, as predicted by the MaxEnt model (Fig 6), far surpassed its current geographic range, illustrating the extensive potential distribution of the species.

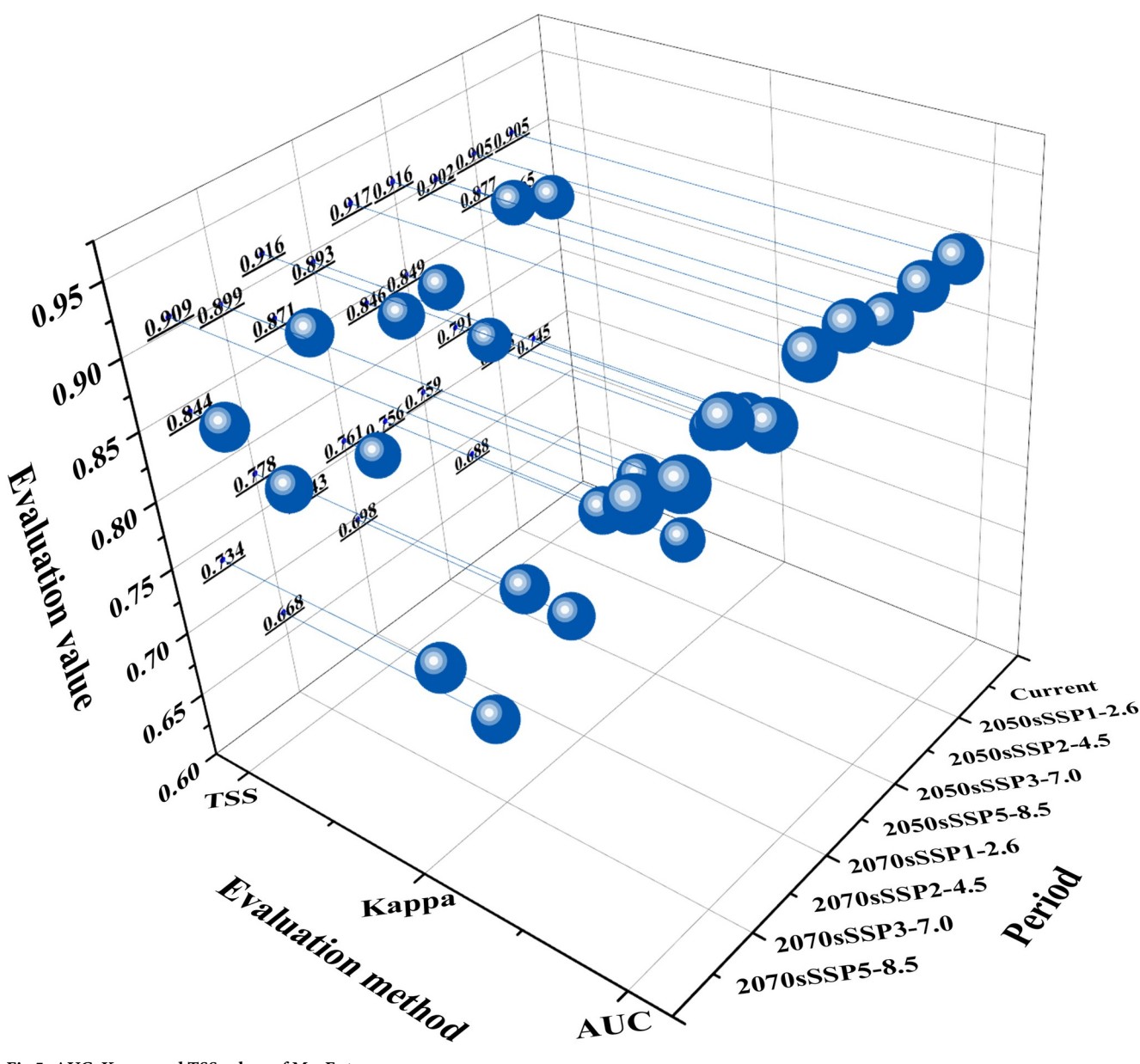

**Fig 5. AUC, Kappa and TSS values of MaxEnt.**

### Future potential geographical distributions of *S. italica*

This study predicted the potentially suitable areas of *S. italica* under four different emission scenarios (SSP1-2.6, SSP2-4.5, SSP3-7.0, and SSP5-8.5) in the 2050s and 2070s, obtaining potentially suitable area maps for *S. italica* under climate change scenarios (Fig 8). Compared to current climate conditions, the suitable area for *S. italica* will significantly decrease in the 2050s and 2070s under all four scenarios. In the 2050s, under the SSP2-4.5 emission scenario, the total suitable area for *S. italica* will decrease by $3.09 \times 10^7$ km$^2$, and in the 2070s, under the same emission scenario, it will decrease by $3.09 \times 10^7$ km$^2$. Under the SSP3-7.0 emission scenario, the total suitable area for *S. italica* will decrease by $3.58 \times 10^7$ km$^2$ in the 2050s and by $3.55 \times 10^7$ km$^2$ in the 2070s. For the SSP1-2.6 emission scenario, the total suitable area for *S.*

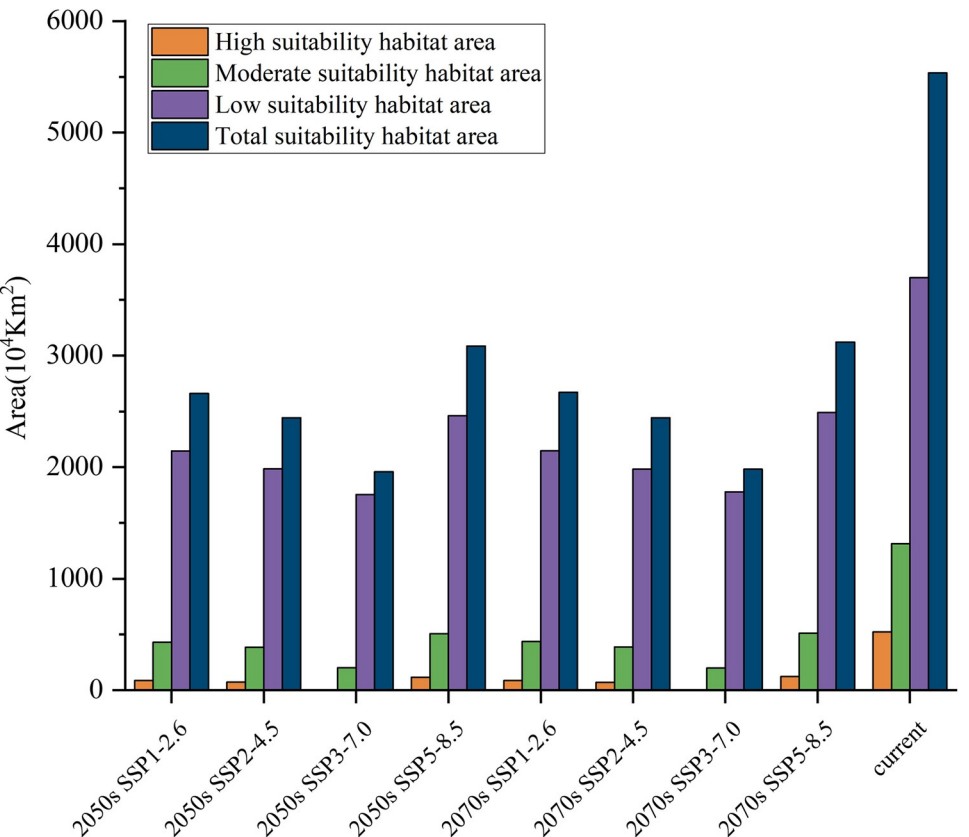

**Fig 6. Suitable areas for *S. italica* under different climate change scenarios($10^4$ km$^2$).**

*italica* will decrease by $2.45 \times 10^7$ km$^2$ in the 2050s and $2.41 \times 10^7$ km$^2$ in the 2070s (Figs 6 and 8). In summary, the most significant reduction in the total suitable area occurs under SSP3-7.0 in both periods, followed by SSP2-4.5, SSP1-2.6, and SSP5-8.5 (Figs 6 and 8).

Under the SSP3-7.0 scenario, our analysis indicates that total suitable areas for *S. italica* will experience a substantial reduction, with a predicted reduction of 64.62% (2050s) and 64.20% (2070s), Highly suitable areas are particularly affected, with a projected decline of 99.51% (2050s) and 99.46% (2070s) in the area of the suitable regions. Moderately suitable areas will also experience a significant reduction, with a decrease of 84.69% (2050s) and 84.74% (2070s). Conversely, lower suitable regions are relatively less affected, with a predicted reduction of 52.56% (2050s) and 51.19% (2070s). Similarly, under the SSP1-2.6 scenario, a significant reduction in suitable habitats is expected, with a decrease of 55.86% (2050s) and 51.74% (2070s). The most suitable habitats will witness a more substantial decline, with a reduction of 83.49% (2050s) and 83.51% (2070s). Medium and low-suitable habitats will also be affected, experiencing cuts of 67.29% and 41.00% in the 2050s and 66.67% and 41.95% in the 2070s, respectively. Under SSP2-4.5, a decrease in total suitable habitats by 44.25% (2050s) and 43.60% (2070s) is expected. Highly suitable habitats will be most affected, with a reduction of 77.56% (2050s) and 76.74% (2070s). Medium and low suitable zones will also decrease by 61.45% and 33.44% in the 2050s and 61.13% and 32.69% in the 2070s, respectively. Lastly, with the SSP5-8.5 scenario, a predicted reduction of 44.25% in total suitable habitats is anticipated in the 2050s and 43.60% in the 2070s. Highly suitable areas will again be most affected, with a decrease of 77.56% (2050s) and 76.74% (2070s). Medium suitable habitats will significantly

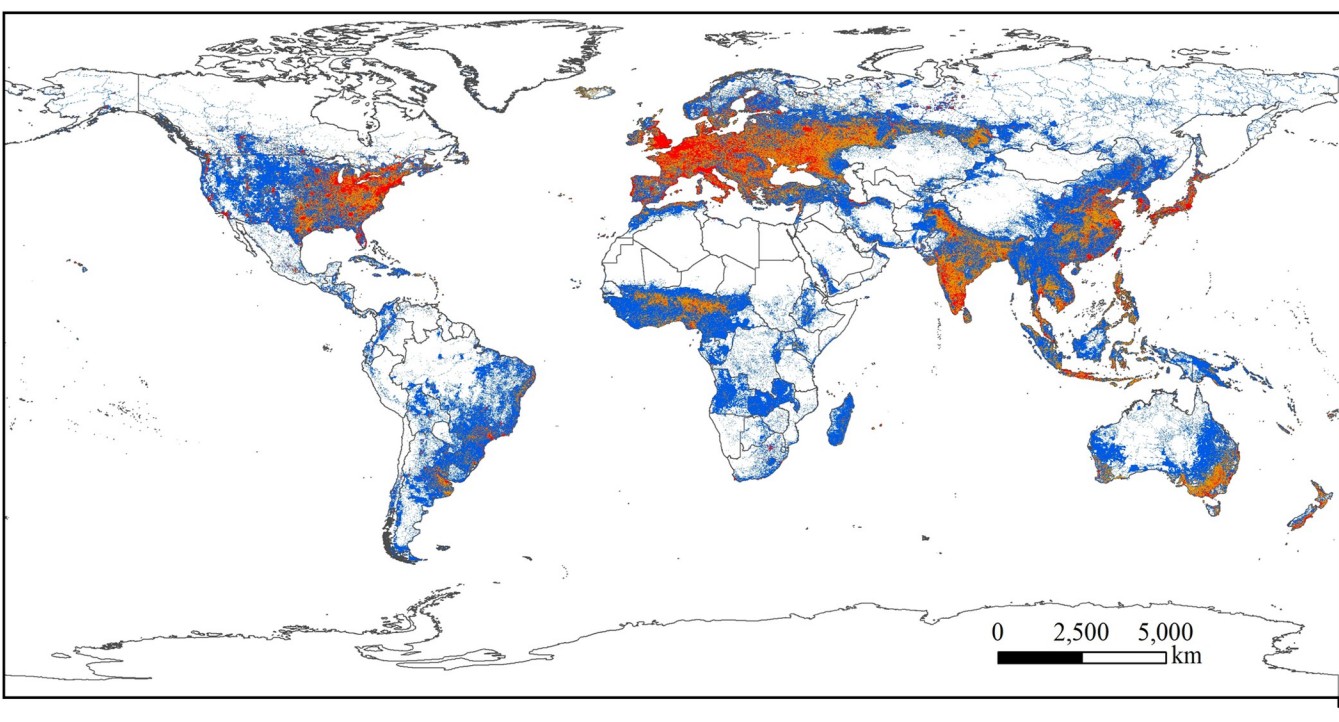

*Setaria italica*  ☐ Unsuitable habitat  ▇ Low suitability habitat  ▇ Modrate suitability habitat  ▇ High suitability habitat

**Fig 7. Current global geographical distributions of *S. italica*.** World administrative maps were acquired from the free National Earth System Science Data Center (http://www.geodata.cn).

decline 61.45% in the 2050s and 61.13% in the 2070s, while lower suitable habitats should be reduced by 33.44% (2050s) and 32.69% (2070s).

In summary, climate change is projected to significantly impact the suitable habitats of *S. italica*, leading to a loss of potential habitats in the coming decades under various emission scenarios. This change is characterized by converting highly suitable areas into medium and low-suitable areas and shifting medium-suitable habitats towards low-suitable and unsuitable habitats (Figs 6 and 8).

## Combinations of dominant environmental variables affecting the distribution of *S. italica*

It is noteworthy that the soil factor, precipitation during the driest month (Bio14), and precipitation during the hottest season (Bio18) consistently ranked among the top three contributors for different periods and different background concentrations of emissions (Fig 9). Consequently, these three factors were utilised as the primary data in this study, offering a focused and robust basis for the subsequent analysis.

The assessment of relationships between the presence probability of *S. italica* and environmental factors was conducted by analysing response curves for each environmental variable. In this analysis, when the presence probability exceeded 0.5, it indicated that the corresponding environmental factor value favoured the growth of *S. italica* [12]. To provide a more intuitive description of the influence of environmental variables on the distribution of *S. italica*, the response curve of the main combinations of environmental variables was derived in this study, as shown in Fig 10. This figure provides insights into how the probability of *S. italica* occurrence varies with the combined values of the selected major environmental factors [12,20].

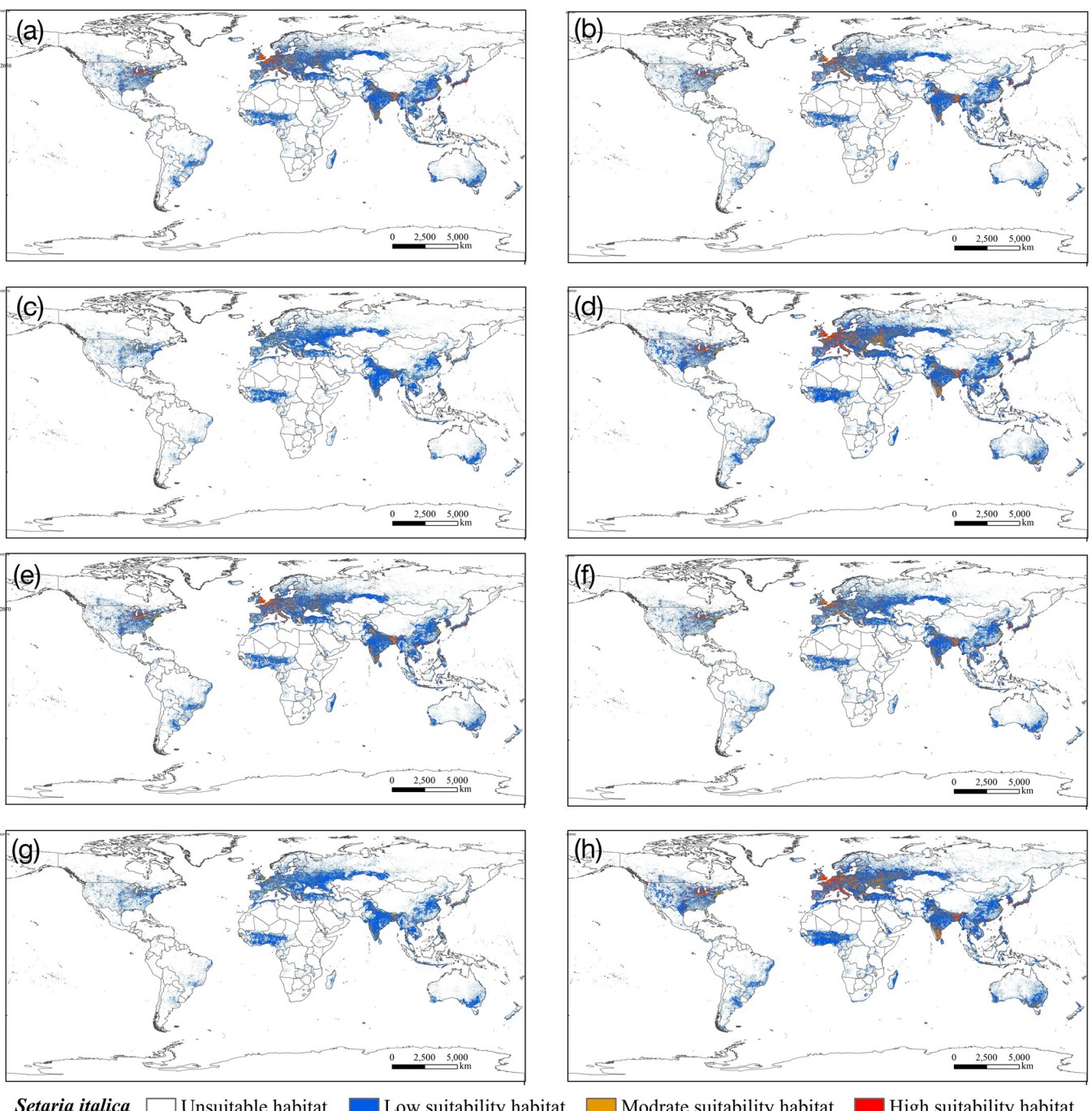

**Fig 8. Potential global geographical distribution of *S. italica* in the 2050s and 2070s.** (a) 2050s, SSP1-2.6; (b) 2050s, SSP2-4.5g; (c) 2050s, SSP3-7.0; (d)2050s SSP5-8.5; (e) 2070s, SSP1-2.6; (f) 2070s, SSP2-4.5g; (g) 2070s, SSP3-7.0; (h)2070s SSP5-8.5. world administrative maps were acquired from the free National Earth System Science Data Center (http://www.geodata.cn).

Based on the results of the MaxEnt model, when the Precipitation of the Coldest Quarter (Bio19) reaches 98.79mm, the survival probability of *S. italica* reaches the threshold for highly suitable habitat (0.5). With a further increase in precipitation in the coldest season (Bio19), the survival probability of *S. italica* peaked at 178.18 mm (0.69). Subsequently, the survival

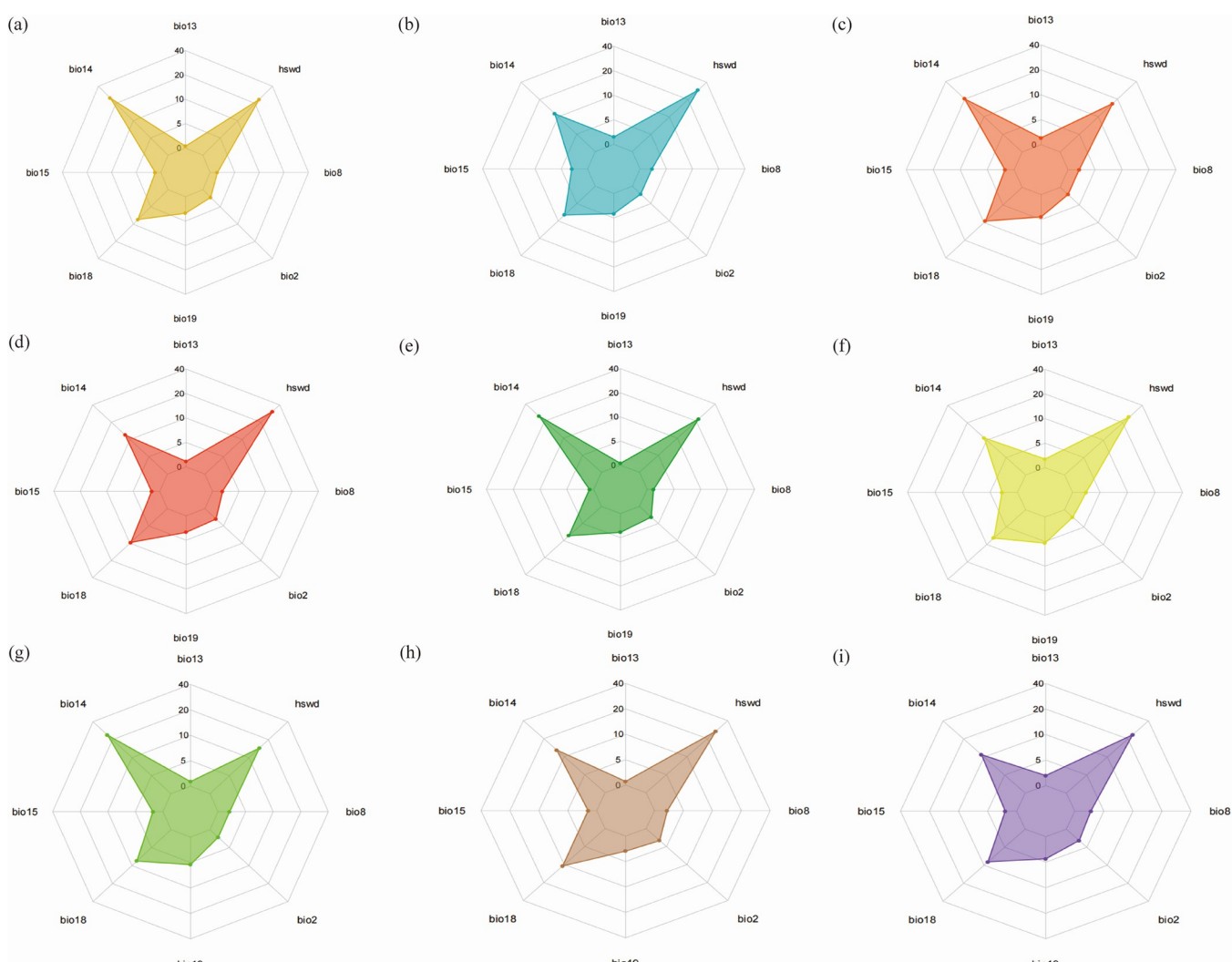

**Fig 9. Environmental variables and their contributions of *S. italica*.** (a) 2050s, SSP1-2.6; (b) 2050s, SSP2-4.5g; (c) 2050s, SSP3-7.0; (d)2050s SSP5-8.5; (e) 2070s, SSP1-2.6; (f) 2070s, SSP2-4.5g; (g) 2070s, SSP3-7.0; (h)2070s SSP5-8.5; (i) current.

probability of *S. italica* gradually decreased with increasing precipitation (Bio19) in the coldest region, reaching 0.5 at 484.89 mm. Thus, the suitable range of Precipitation of the Coldest Quarter (Bio19) for *S. italica* survival is 98.79mm to 484.89mm. Similarly, when the Precipitation of the Warmest Quarter (Bio18) reaches 145.89mm, the survival probability of *S. italica* reaches the threshold for highly suitable habitat (0.5). The survival probability of *S. italica* peaks (0.61) at 174.28mm of Precipitation in the Warmest Quarter (Bio18) and then decreases with increasing precipitation in the warmest quarter (Bio18), reaching 0.5 at 314.92 mm. Therefore, the suitable range of Precipitation of the Warmest Quarter (Bio18) for *S. italica* survival is 145.89mm to 314.92mm. Furthermore, when Precipitation of Driest Month (Bio14) reaches 22.96mm, the survival probability of *S. italica* reaches the threshold for highly suitable habitat (0.5). The survival probability of *S. italica* peaks (0.70) at 45.63mm of Precipitation of Driest Month (Bio14), declining to 0.5 at 102.05mm. Consequently, the suitable range of Precipitation of Driest Month (Bio14) for S. italica survival is 22.96mm to 102.05mm. In summary, *S. italica* exhibits a relatively narrow suitable range for Precipitation of Driest Month (Bio14), indicating its high requirement for precipitation.

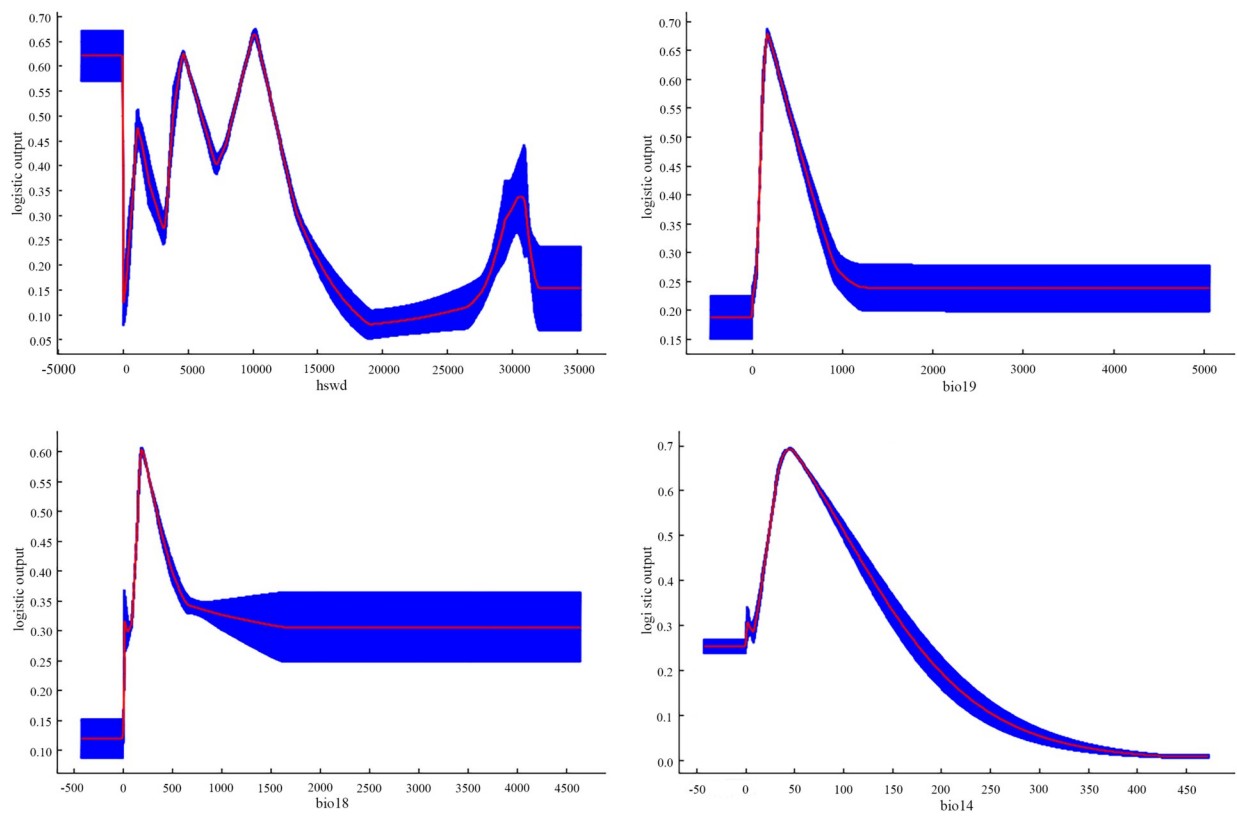

**Fig 10. Response curves of existence probability of *S. italica*.**

## Changes in the potential geographical distribution of *S. italica* under climate change scenarios

The suitable habitats of S. italica in the 2050s and 2070s exhibit a trend of contraction to the west in Europe, to the north in Africa, and to the south in Asia, North America, South America, and Oceania. In the 2070s, the SSP5-8.5 emission scenario indicates the smallest loss of suitable habitats of S. italica, with an area of 2.37 × 107 km2 (Fig 11). Conversely, the most significant loss of suitable habitat occurs in the 2050s under the SSP3-7.0 emission scenario, with an area of 3.58 × 107 km2 (Fig 11). China, the United States, Brazil, and Australia are the areas with the most significant losses.

## Discussion

### Changes in the potential geographic distribution of *S. italica* under future climate change scenarios

Based on environmental factors under four emission scenarios in 2050 and 2070, combined with contemporary climate conditions, the MaxEnt model was employed to predict the potential geographic distribution of *S. italica* in China under future climate change scenarios. Spatial overlay analysis of the results was conducted in ArcGIS to illustrate the potential distribution changes of *S. italica* under future climate change scenarios (Fig 11). The model projections indicate that the potential geographic distribution of *S. italica* in the 2050s and 2070s under the four emission scenarios is projected to be lower than the potential distribution under modern climate conditions.

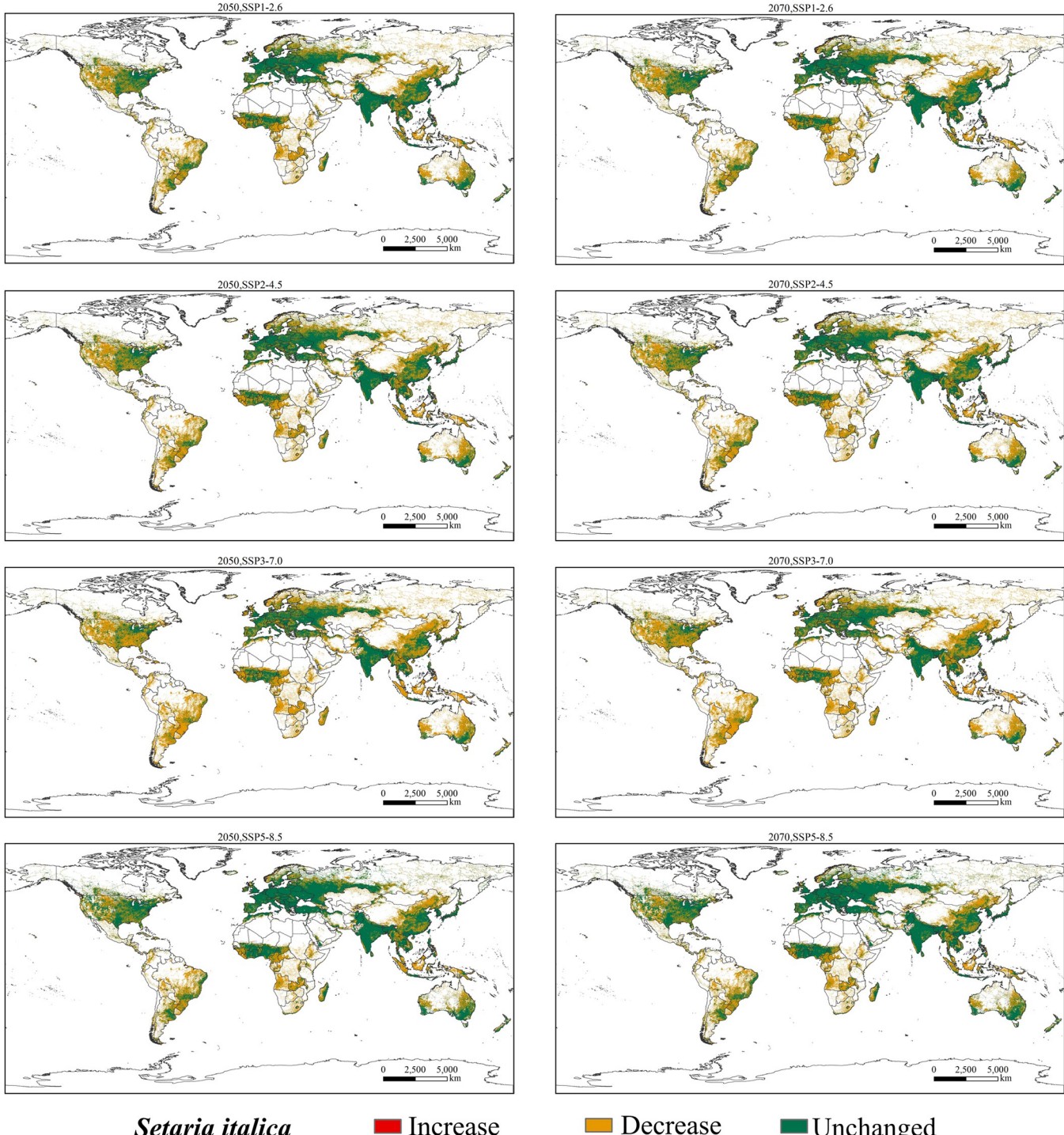

**Fig 11. Changes in the potential geographical distribution of *S. italica* under climate change scenarios.** In the future. World administrative maps were acquired from the free National Earth System Science Data Center (http://www.geodata.cn).

In many areas, the suitability of *S. italica* shows an increasing and then decreasing trend, leading to a decrease in the probability of survival, especially in the SSP3-7.0 emission scenario, where temperature and precipitation values are outside the suitability range. This is consistent

with findings from Thomas, who studied the extinction risk of organisms on a sample area covering 20% of the Earth's surface, suggesting varying impacts on different species in response to climate change [35]. While some species benefit from climate warming, the effect on *S. italica* is mainly negative.

Under the SSP5-8.5 emission scenario, *S. italica* is projected to experience the least reduction in suitable habitats at all levels. This resilience is attributed to the species' ability to tolerate high temperatures and drought, making it less vulnerable to this particular emission scenario. These results align with other studies examining the response of various species to changing climate conditions.

It's worth noting that climate change could indirectly affect the population and distribution characteristics of *S. italica* by directly influencing the ecological environment [36]. Additionally, irrational human activities, such as urban construction, hydroelectric development, and other industrial practices, could further contribute to a dramatic decline in potential distribution areas of *S. italica* [37]. This study focused on environmental factor variables for two periods (2050 and 2070). In future studies, multiple periods should be considered to derive an overall trend in the potential geographic distribution of the species in response to climate change.

## Constraints of environmental variables on the potential geographical distribution of *S. italica*

The MaxEnt model results indicate that soil and precipitation factors, specifically the driest monthly and hottest seasonal precipitation, are critical environmental factors limiting the potential geographic distribution of *S. italica*. The study reveals that variations in these precipitation factors influence the probability of *S. italica* presence, with increased driest-month precipitation and hottest-season precipitation positively impacting the probability of existence. This is consistent with previous research by Zhang and Yang [36,38], demonstrating the sensitivity of *S. italica* growth to low and excessive precipitation levels. The study confirms Zhang's findings by emphasising the significant impact of moisture conditions on *S. italica* production, and it is expected that increased precipitation will significantly impact the probability of this species' existence in the future [39]. Zhang's study emphasises the significant impact of precipitation on the yield of *S. italica*. The research indicates that with increased precipitation, the species' survival probability will first rise. However, as precipitation continues to grow to a certain extent, the survival probability of *S. italica* decreases.

Based on actual weather station data, Cao's study supports the conclusion that climatic variables such as the driest monthly precipitation and hottest seasonal precipitation play a primary role in constraining the growth and distribution of *S. italica* [40]. These results are consistent with the findings of this study. Further, the influence of soil factors on *S. italica* growth and development is supported by studies of Kaur, Bandyopadhyay, and Nissi, highlighting the significance of soil fertility and microorganisms in soil as critical environmental factors for *S. italica* [41–43].

Historical studies by Dong and Chen, exploring the development and decline of *S. italica* cultivation over millennia, revealed the importance of temperature as a significant constraint on the potential geographical distribution of *S. italica* [2,9]. These findings also highlight that temperature factors significantly constrain the potential geographic distribution of *S. italica*.

The study focuses on predicting the potential geographic distribution of *S. italica* in China and identifies the environmental variables limiting its potential distribution. Over time, changes in the study area may impact the environmental factors influencing *S. italica* growth. Although other environmental factors, such as vegetation cover, could affect the species'

potential distribution, they were not included in the study due to challenges in accurately predicting global vegetation cover in the future. Therefore, local hydrogeological conditions should be considered when applying the potential geographic distribution areas derived from this study in agricultural production. Overall, the study contributes valuable insights to the global macro plan for the rational cultivation of *S. italica*.

### The importance of conducting simulations of the potential geographic distribution of *S. italica*

Global climate change seriously threatens agricultural production, directly impacting food security and, consequently, human development. The food crisis triggered by climate change is far-reaching and has become one of the most severe challenges globally. This study used ecological niche modelling to predict the potential geographic distribution of *S. italica*, which contributes to a clearer understanding of how this crop responds to climate change. As a significant global miscellaneous crop, *S. italica* is crucial in ensuring food security and human health.

## Conclusions

The simulation of the potential geographical distribution of *S. italica* based on current climate conditions reveals its main distribution areas, including the United States (North America), Brazil (South America), Australia (Oceania), China, India (Asia), and the Russian Federation (Europe). However, under future climate change scenarios, there is a significant reduction in the potential distribution areas of *S. italica*, which could have unfavourable implications for its cultivation. Critical environmental factors limiting the potential geographical distribution of *S. italica* include soil and precipitation factors, particularly the driest monthly and hottest seasonal precipitation. In conclusion, the study suggests that climate change has the potential to significantly impact the future distribution pattern of *S. italica* cultivation, leading to a reshaping of production and trade patterns. These predicted results serve as a crucial step in the macro plan for the rational cultivation of *S. italica*. To address these challenges, it is recommended that the cultivation of *S. italica* be promoted in highly suitable areas, especially in the context of grassland restoration. Increasing knowledge and implementing practices for controlling pests and diseases affecting *S. italica* are essential. Immediate actions, such as quarantine measures for affected crops and treatment of diseases and insect infestations, are crucial. By applying scientific knowledge and effective control measures, it is possible to mitigate the impact of climate change on *S. italica* cultivation, contributing to broader food security efforts.

## Supporting information

**S1 Table. *Setaria italica* occurrence records data.**
(XLS)

**S2 Table. Pairwise Pearson's correlation coefficients of environmental variables.**
(XLSX)

## Acknowledgments

We sincerely thank Zhangqiang You in the Ecological Security and Protection Key Laboratory of Sichuan Province for providing us with technical support in this study.

## Author Contributions

**Conceptualization:** Yi Huang.

**Data curation:** Xue Jiang, Mei Liu, Zixi Shama, Jiayi Li.

**Software:** Jingtian Yang, Yi Huang.

**Writing – original draft:** Jingtian Yang, Yunlong Ma.

**Writing – review & editing:** Yi Huang.

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
