## [Decision Letter · Decision Letter 0]

18 Jan 2024

PONE-D-23-42450Potential global distribution of Setaria italica , an important species for dryland agriculture in the context of climate changePLOS ONE

Dear Dr. Huang,

Thank you for submitting your manuscript to PLOS ONE. After careful consideration, we feel that it has merit but does not fully meet PLOS ONE’s publication criteria as it currently stands. Therefore, we invite you to submit a revised version of the manuscript that addresses the points raised during the review process.

We look forward to receiving your revised manuscript.

Kind regards,

Jiban Shrestha

Academic Editor

PLOS ONE

Journal Requirements:

4. Please note that funding information should not appear in any section or other areas of your manuscript. We will only publish funding information present in the Funding Statement section of the online submission form. Please remove any funding-related text from the manuscript.

   "The Scientific research initiation project of Mianyang Normal University (QD2019A13, QD2021A37 and QD2023A01), the Funding of the Open Project from the Ecological Security and Protection Key Laboratory of Sichuan Province (ESP1608, ESP2008, ESP2201 and ESP2204), the Sichuan Provincial Education Department Scientific Research Project (15ZB0283), the Sichuan Provincial Science and Technology Department Project (2023NSFSC0750)."

6. We note that your Data Availability Statement is currently as follows: All relevant data are within the manuscript and its Supporting Information files.

7. PLOS requires an ORCID iD for the corresponding author in Editorial Manager on papers submitted after December 6th, 2016. Please ensure that you have an ORCID iD and that it is validated in Editorial Manager. To do this, go to ‘Update my Information’ (in the upper left-hand corner of the main menu), and click on the Fetch/Validate link next to the ORCID field. This will take you to the ORCID site and allow you to create a new iD or authenticate a pre-existing iD in Editorial Manager. Please see the following video for instructions on linking an ORCID iD to your Editorial Manager account: https://www.youtube.com/watch?v=_xcclfuvtxQ

8. We note that Figures 3, 6, 7 and 11 in your submission contain map/satellite images which may be copyrighted. All PLOS content is published under the Creative Commons Attribution License (CC BY 4.0), which means that the manuscript, images, and Supporting Information files will be freely available online, and any third party is permitted to access, download, copy, distribute, and use these materials in any way, even commercially, with proper attribution. For these reasons, we cannot publish previously copyrighted maps or satellite images created using proprietary data, such as Google software (Google Maps, Street View, and Earth). For more information, see our copyright guidelines: http://journals.plos.org/plosone/s/licenses-and-copyright.

a. You may seek permission from the original copyright holder of Figures 3, 6, 7 and 11 to publish the content specifically under the CC BY 4.0 license.  

Additional Editor Comments:

The authors need to include suggestions given by both reviewers before re-submission.

Reviewers' comments:

Reviewer's Responses to Questions

**Comments to the Author**

1. Is the manuscript technically sound, and do the data support the conclusions?

Reviewer #1: Yes

Reviewer #2: Partly

2. Has the statistical analysis been performed appropriately and rigorously? 

Reviewer #1: Yes

Reviewer #2: N/A

3. Have the authors made all data underlying the findings in their manuscript fully available?

Reviewer #1: Yes

Reviewer #2: Yes

4. Is the manuscript presented in an intelligible fashion and written in standard English?

Reviewer #1: No

Reviewer #2: Yes

5. Review Comments to the Author

Reviewer #1: Major modifications:

1.All the images are all very blurry, please increase their resolution。

2.2.In Abstract, ①Abstract transition too long.②the author evaluated the prediction accuracy of the Maxent model using three approaches: AUC values, TSS values, and Kappa statistics, please add numerical values. The number should be changed to expression such as 5.54×107. ③Compared with the modern potential distribution, we predict that the four future climate change scenarios will lead to varying degrees of reduction in the likely geographic ranges of S. italica. Please add numerical values and then Draw a conclusion. ⑷which could reshape the global S. italica production and trade patterns. Please give specific measures.

3.Over ten distinct models, including Bioclim, Domain, GARP, MaxEnt, and others. Please provide the full name of the abbreviation, for example, MaxEnt(Maximum Entropy Model).......

4.Climate and digital elevation data for both current and future emission scenarios were retrieved from the WorldClim database......... Please provide a detailed description, such as 19 bioclimatic variables with 30-arc-second (ca.1km2 at ground level) spatial resolution, under the current (i.e., in the period of 1970-2000) condition......, the altitude, Soil data and UV-B radiation data remain the same under the current and future condition?

5.we conducted a Spearman correlation analysis as depicted in Figure 4........ Please provide detailed description of the analysis process, such as firstly extract vaules and then Spearman correlation analysis........

6.2.3 Model construction and evaluation. Please point out the source of The MaxEnt modeling.

7.The accuracy of the evaluated model was categorized into five levels: failure,....... Please provide the cited references.

8.Based on the results (Fig. 5), .......... Please provide specific numerical values so that readers can compare them with the above standards provided by author.

9.when the presence probability surpassed 0.5( add to cited references).

And give the thresholds of these dominant environmental variables.

10.Combine 4.3 and Conclusion into a paragraph.

11.In reference, the literature format does not meet the requirements of this journal.

12.Figure11, Changes in the potential geographical distribution of S. italica under climate change scenarios in the future. Please add to research method on how to analysis and draw.

13.4.2 Constraints of environmental variables on the potential geographical distribution of S. italica. Please add to the thresholds of these dominant factors and discuss the research result based on them.

14. Unfortunately, the paper is very hard to read and requires significant rewriting and restructuring before publication. I strongly recommend that the author invite some editing companies to polish the article for readability.

15.Please add to the contributions of all authors.

16.3.3 Future potential geographical distributions of S. italica.This section should be described as a whole, not separately, and finally summarized

Reviewer #2: The methods are not well described in detail.

The discussion part should be explained more and further.

The language should be checked by a native English speaker.

Please see the comments in the attached file.

6. PLOS authors have the option to publish the peer review history of their article (what does this mean?). If published, this will include your full peer review and any attached files.

Reviewer #1: No

Reviewer #2: No

---

## [Author Response · Author response to Decision Letter 0]

20 Mar 2024

Dear Editors and Reviewers:

Thank you for your letter and for the reviewers’ comments concerning our manuscript entitled “Potential global distribution of Setaria italica, an important species for dryland agriculture in the context of climate change” (ID: PONE-D-23-42450). Those comments are all valuable and very helpful for revising and improving our paper, as well as the important guiding significance to our research. We have studied comments carefully and have made corrections which we hope meet with approval. Those changes are marked with red within the manuscript. The revision notes are given as follows:

Responds to the Academic editor and reviewer’s comments:

Academic editor：

Q1：Please ensure that your manuscript meets PLOS ONE's style requirements, including those for file naming. The PLOS ONE style templates can be found at 

Response: Thank you for reminding us to adhere to PLOS ONE's style requirements, including those for file naming. We have already made the necessary adjustments to ensure compliance.

Q2: We suggest you thoroughly copyedit your manuscript for language usage, spelling, and grammar. If you do not know anyone who can help you do this, you may wish to consider employing a professional scientific editing service.  

Response: We tried our best to improve the language in the revised manuscript. we do invite dr. Cong Zhao, who’s a well established expert, to polish our paper. Please see if the revised version met the English presentation standard.

Q3: In your Methods section, please provide additional information regarding the permits you obtained for the work. Please ensure you have included the full name of the authority that approved the field site access and, if no permits were required, a brief statement explaining why.

Response: Thank you for your feedback. We have ensured that we included the full name of the authority that approved the field site access in the Methods section. This authority provided the necessary permits for conducting our research, allowing us to access the field sites mentioned in the manuscript.

Q4: Please note that funding information should not appear in any section or other areas of your manuscript. We will only publish funding information present in the Funding Statement section of the online submission form. Please remove any funding-related text from the manuscript.

Response: Thank you for reminder, we have deleted funding information in our manuscript

Q5: Thank you for stating the following financial disclosure: 

 "The Scientific research initiation project of Mianyang Normal University (QD2019A13, QD2021A37 and QD2023A01), the Funding of the Open Project from the Ecological Security and Protection Key Laboratory of Sichuan Province (ESP1608, ESP2008, ESP2201 and ESP2204), the Sichuan Provincial Education Department Scientific Research Project (15ZB0283), the Sichuan Provincial Science and Technology Department Project (2023NSFSC0750)."

Response: Thank you for your feedback. We have revised the Funding Statement to include all sources of support received during this study, as detailed in your guide for authors. The amended statement is provided below:

"This work was supported by grants from The Scientific research initiation project of Mianyang Normal University (QD2019A13, QD2021A37 and QD2023A01; URL: https://www.mtc.edu.cn/), the Funding of the Open Project from the Ecological Security and Protection Key Laboratory of Sichuan Province (ESP1608, ESP2008, ESP2201 and ESP2204; URL: https://zdsys.mtc.edu.cn/), the Sichuan Provincial Education Department Scientific Research Project (15ZB0283; URL:http://edu.sc.gov.cn/), the Sichuan Provincial Science and Technology Department Project (2023NSFSC0750; URL: https://kjt.sc.gov.cn/). The funders had a role in study design, data collection and analysis, decision to publish, or preparation of the manuscript. There was no additional external funding received for this study."

We have also included the amended Funding Statement within our cover letter as requested.

Thank you for your assistance in updating the online submission form.

Q6: We note that your Data Availability Statement is currently as follows: All relevant data are within the manuscript and its Supporting Information files.

Response: Thank you for your message. We confirm that our submission contains all the raw data required to replicate the results of our study. 

Q7: PLOS requires an ORCID iD for the corresponding author in Editorial Manager on papers submitted after December 6th, 2016. Please ensure that you have an ORCID iD and that it is validated in Editorial Manager. To do this, go to ‘Update my Information’ (in the upper left-hand corner of the main menu), and click on the Fetch/Validate link next to the ORCID field. This will take you to the ORCID site and allow you to create a new iD or authenticate a pre-existing iD in Editorial Manager. Please see the following video for instructions on linking an ORCID iD to your Editorial Manager account: https://www.youtube.com/watch?v=_xcclfuvtxQ

Response: Thank you for informing us about the requirement for an ORCID iD for the corresponding author in Editorial Manager. I have ensured that I have an ORCID iD and that it is validated in Editorial Manager. I have followed the instructions provided and have linked my ORCID iD to my Editorial Manager account. 

Q8: We note that Figures 3, 6, 7 and 11 in your submission contain map/satellite images which may be copyrighted. All PLOS content is published under the Creative Commons Attribution License (CC BY 4.0), which means that the manuscript, images, and Supporting Information files will be freely available online, and any third party is permitted to access, download, copy, distribute, and use these materials in any way, even commercially, with proper attribution. For these reasons, we cannot publish previously copyrighted maps or satellite images created using proprietary data, such as Google software (Google Maps, Street View, and Earth). For more information, see our copyright guidelines: http://journals.plos.org/plosone/s/licenses-and-copyright.

 a. You may seek permission from the original copyright holder of Figures 3, 6, 7 and 11 to publish the content specifically under the CC BY 4.0 license.  

Response: Thank you for bringing this to our attention. world administrative maps were acquired from the free National Earth System Science Data Center (http://www.geodata.cn), copyright (World administrative maps were acquired from the free National Earth System Science Data Center (http://www.geodata.cn)) have been added to the legends of figures 3, 7, 8, and 11.

Reviewer1：

Q1: All the images are all very blurry, please increase their resolution.

Response: Thank you for your reminder, we have separately uploaded high-resolution versions of the images. The blurriness of the images may have occurred during the PDF file generation process.

Q2: In Abstract, ①Abstract transition too long.②the author evaluated the prediction accuracy of the Maxent model using three approaches: AUC values, TSS values, and Kappa statistics, please add numerical values. The number should be changed to expression such as 5.54×107. ③Compared with the modern potential distribution, we predict that the four future climate change scenarios will lead to varying degrees of reduction in the likely geographic ranges of S. italica. Please add numerical values and then Draw a conclusion. 

④which could reshape the global S. italica production and trade patterns. Please give specific measures.

Response: Thank you for your suggestions. Regarding ①, the revisions have been made, removing redundant parts; Regarding ②, specific numerical values have been added in Figure 5. The numerical expression has been changed to X×107; Regarding ③, specific numerical values for the reduction of each suitability zone have been added in the Results. In the subsequent paragraphs, percentage descriptions of the changes in area for each suitability zone in various periods have been provided(Lines208-218); Regarding④, We have re-written the sentence(Lines32-33).

Q3. Over ten distinct models, including Bioclim, Domain, GARP, MaxEnt, and others. Please provide the full name of the abbreviation, for example, MaxEnt(Maximum Entropy Model).......

Response: Thank you for your suggestions, The full names have been added after each model abbreviation(Lines55-56).

Q4. Climate and digital elevation data for both current and future emission scenarios were retrieved from the WorldClim database......... Please provide a detailed description, such as 19 bioclimatic variables with 30-arc-second (ca.1km2 at ground level) spatial resolution, under the current (i.e., in the period of 1970-2000) condition......, the altitude, Soil data and UV-B radiation data remain the same under the current and future condition?

Response: Thank you for your suggestions, First, this study was conducted under the condition where all environmental factors except climatic variables remained constant. Second, a more detailed description was provided regarding the acquisition of environmental variables.

We obtained 19 climatic environmental datasets with a coordinate system of WGS84 and a grid size of 2.5 arc-minutes from the WorldClim database (http:// www.world-clim.org//). This database collected detailed meteorological information from meteorological stations worldwide from 1970 to 2000. Future climate data were derived from the BCC-CSM2-MR climate system model developed by the National Climate Center, covering four emission scenarios. Soil data were sourced from the Harmonized World Soil Database (HWSD V1.2, https://www.fao.org/). UV-B radiation data were obtained from the global UV-B download radiation dataset (gIUV, https://www.ufz.de/gluv/index.php). The world administrative map was downloaded from the National Geospatial Information System Science Data Center of China (http://www.geodata.cn). ArcGIS software (version 10.2) was used to standardize the spatial resolution of all environmental variables. Referring to the methods of ZHAO et al. [15] and Jing et al. [8], the resolution of all environmental variables was set to 5 arc-minutes.

Due to the correlations among environmental variables, it was necessary to conduct a correlation analysis before applying the MaxEnt model. In this study, Spearman correlation analysis was performed on environmental factors, and the correlation analysis results are shown in Figure 4. When the correlation coefficient between two environmental factors was greater than or equal to 0.8, the one with a higher contribution rate was retained. Ultimately, eight environmental factors were selected for the operation of the MaxEnt model(Lines116-125).

Q5. we conducted a Spearman correlation analysis as depicted in Figure 4........ Please provide detailed description of the analysis process, such as firstly extract vaules and then Spearman correlation analysis........

Response: Thank you for your suggestions, The process of filtering all environmental variables has been detailed according to the filtering steps.

Due to the correlation among environmental variables, it was necessary to conduct a correlation analysis before applying the MaxEnt model. The selection of environmental factors was conducted in two steps: (1) Initially, all environmental factors were imported into the MaxEnt model for computation three times, and ecological factors with zero contribution rates were removed; (2) Subsequently, all environmental factors with contribution rates greater than 0 were selected for Spearman correlation analysis. When the correlation coefficient between two environmental factors was ≥ 0.8, the one with a higher contribution rate was retained. Ultimately, eight environmental factors were selected for the operation of the MaxEnt model (Fig 4)(Lines130 -136).

Q6. 2.3 Model construction and evaluation. Please point out the source of The MaxEnt modeling.

Response: Thank you for your suggestions, The software versions and sources used in model construction have been provided. MaxEnt version 3.4.1 was downloaded from the website (http://biodiversityinformatics.amnh.org/open source/MaxEnt/). The ArcGIS software version used in this study was version 10.2(Lines128, 139).

Q7. The accuracy of the evaluated model was categorized into five levels: failure,....... Please provide the cited references.

Response: Thank you for your suggestions, References have been added(Lines153).

Q8. Based on the results (Fig. 5), .......... Please provide specific numerical values so that readers can compare them with the above standards provided by author.

Response: Thank you for your suggestions, Specific numerical values have been added to Fig 5.

Q9. when the presence probability surpassed 0.5( add to cited references).And give the thresholds of these dominant environmental variables.

Response: Thank you for your suggestions, References and the thresholds of these dominant environmental variables have been added(Lines258, 264-280).

Q10. Combine 4.3 and Conclusion into a paragraph.

Response: Thank you for your suggestions, The content of section 4.3 has been integrated into the Conclusion(Lines362-377).

Q11. In reference, the literature format does not meet the requirements of this journal.

Response: Thank you for reminding us the literature format does not meet the requirements of this journal, We have already made the necessary adjustments to ensure compliance(Lines382-513).

Q12. Figure11, Changes in the potential geographical distribution of S. italica under climate change scenarios in the future. Please add to research method on how to analysis and draw.

Response: Thank you for your suggestions, We have added additional details on the research method for analyzing and drawing Figure 11(Lines179-184).

Q13. 4.2 Constraints of environmental variables on the potential geographical distribution of S. italica. Please add to the thresholds of these dominant factors and discuss the research result based on them.

Response: Thank you for your suggestions, the thresholds of these dominant environmental variables have been added and discussed(Lines264-280).

Q14. Unfortunately, the paper is very hard to read and requires significant rewriting and restructuring before publication. I strongly recommend that the author invite some editing companies to polish the article for readability.

Response: Thank you for your suggestions, We tried our best to improve the language in the revised manuscript. we do invite dr. Cong Zhao, who’s a well established expert, to polish our paper. Please see if the revised version met the English presentation standard.

Q15. Please add to the contributions of all authors.

Response: Thank you for your suggestions, All authors' contributions have been checked and marked in the Contributor Roles section of the submission system.

Q16. 3.3 Future potential geographical distributions of S. italica.This section should be described as a whole, not separately, and finally summarized

Response: Thank you for your suggestions, in Results, the changes in its potential distribution have already been described as a whole and and finally summarized(Lines208-245).

Reviewer2：

Q1.The methods are not well described in detail.

Response: Thank you for your suggestions, in Materials and methods, We have already rewritten and improved it.

Q2.The discussion part should be explained more and further.

Response: Thank you for your suggestions, in Materials and methods, We have already rewritten and improved it.

Q3.The language should be checked by a native English speaker.

Response: Thank you for your suggestions, We tried our best to improve the language in the revised manuscript. we do invite dr. Cong Zhao, who’s a well established expert, to polish our paper. Please see if the revised version met the English presentation standard.

the comments in the attached file：

Q1. I suggest the aims should be separately listed in this part.

Response: Thank you for your suggestions, The objectives of this study have been separately listed (lines92-96).

Q2. How do you deal with your initial dataset collected from several databases? This is crucial because it determines the accuracy and effectiveness of your SDMs next. 

Response: Thank you for your suggestions, We have improved the process of collecting the initial dataset from several databases. To obtain the occurrences of S. italica, we conducted field surveys and made queries in databases such as the Global Biodiversity Database (http://www.gbif.org/, last accessed on April 2023), China Plantwise (http://www.iplant.cn, last accessed on April 2023) and China Digital Herbarium (https://www.cvh.ac.cn, last accessed on April 2023) [15, 18]. Coordinates for S. italica were confirmed and screened to eliminate records with insufficient or redundant descriptions. To avoid overfitting of the model caused by concentrated distribution data in certain regions, which could introduce uncertainty, the Buffer tool in ArcGIS 10.2 was utilized in this study. Following the resolution of environmental variables, a buffer zone with a radius of 5 km was established around each distribution point, allowing for only one distribution point within 5 km. In the end, 3154 distribution points of S. italica were collected, and a distribution map (Fig 3) was generated. All records were imported into Microsoft Excel 2020 and saved in "CSV" format(lines102-112).

Q3. Which versions, GCMs, resolution do you apply? Please make details for this contents. 

Response: Thank you for your suggestions, we have improved the details for this content, We have provided the sources of environmental factors and the software version used. For our study, we gathered essential data from reputable sources to conduct a comprehensive analysis of the potential distribution of S. italica. Future climate data for the 2050s (2041–2060) and 2070s (2061–2080) under four representative concentration pathways (SSP1-2.6, SSP2-4.5, SSP3-7.0, and SSP5-8.5) were generated using the Community Climate System Model (GCSM) 4.0. Climate and digital elevation data for both current and future emission scenarios were retrieved from the WorldClim database (DEM, http://www.worldclim.org//, last accessed on April 2023). Soil data, integral to our investigation, corresponds to the Coordinated World Soil Database (HWSD V1.2, https://www.fao.org/, last accessed on April 2023), ensuring a robust understanding of the soil characteristics. UV-B radiation data, a crucial environmental factor, was sourced from the global UV-B radiation dataset (gIUV, https://www.ufz.de/gluv/index.php, last accessed on April 2023). 

ArcGIS software (version 10.2) was used to standardize the spatial resolution of all environmental variables. Following the methods of ZHAO et al. [15] and Jing et al. [8], the resolution of all environmental variables was set to 5 arc-minutes. The MaxEnt model was implemented using version 3.4.1, downloaded from the website (http://biodiversityinformatics.amnh.org/open source/MaxEnt/)(lines116-140).

Q4. Considering

Response: Thank you for your suggestions, we have corrected(line130). 

Q5. How do you set other parameters of MaxEnt models? Please provide a detailed explanation.

Response: Thank you for your suggestions, We have already described this in the methods section.

In this study, we implemented trial models with different parameter settings to evaluate performance, optimizing these models based on the observed response to known distribution points of S. italica and their corresponding environmental factors. The adjustment process involved establishing the range of regularization multipliers (RM) from 0.5 to 4. Additionally, six feature combinations (FC) were employed to optimize the model parameters: L (linear features), LQ (linear features + quadratic features), H (hinge features), LQH (linear features + quadratic features + hinge features), LQHP (linear features + quadratic features + hinge features + product features), and LQHPT (linear features + quadratic features + hinge features + product features + threshold features). After careful evaluation, the optimal combination for the regularization multiplier and feature selection was determined to be RM and LQHPT, respectively(lines154-163).

Q6.The section "Commonly.....very good" should be swapped with the following section.

Response: Thank you for your suggestions, We have swapped with the following section(Line187). 

Q7.Results

Response: Thank you for your suggestions, we have corrected(Line185). 

Q8.You mention the "very good" rather than "excellent" in the above parts.

Response: Thank you for your suggestions, we have corrected(Line187). 

Q9.Please express the detailed information of the location of highly suitable area.

Response: Thank you for your suggestions, we have added detailed information on the location of a highly suitable area(Lines196-197). 

Q10. Area

Response: Thank you for your suggestions, we have corrected(Line200). 

Q11. Please re-rank the location of the figures of your manuscript. 

Response: Thank you for your suggestions, we have re-rank the location of the figures of our manuscript. 

Q12. Please rewrite the sentence

Response: Thank you for your suggestions, We have re-written the sentence(Lines202-203). 

Q13. Please rewrite the sentence.

Response: Thank you for your suggestions, We have re-written the sentence(Line220). 

Q14. This part should be incorporate into Material and Methods.

Response: Thank you for your suggestions, We have incorporated this part into Material and Methods.(Lines165-171).

Q15. Please rewrite the sentence.

Response: Thank you for your suggestions, We have re-written the sentence(Lines298-300). 

Q16. I do not know what you mean, please rewrite the sentence.

Response: Thank you for your suggestions, We have re-written the sentence(Lines303-304). 

Q17. Clearly, the meaning of this sentence do not support the results obtained by your manuscript, and this sentence sounds some vague. Please rewrite the sentence. 

Response: Thank you for your suggestions, We have re-written the sentence(Lines305-307). 

Q18. Thomas et al. (???)

Response: Thank you for your suggestions, We have re-written the sentence(Line306).

Q19. This part should be incorporated into Results sections.

Response: Thank you for your suggestions, We have incorporated this part into the Results sections(Lines284-292).

Q20. Which study do you align with? Actually, I do not see any in-depth discussion in your manuscript.

Response: Thank you for your suggestions, We have re-written the sentence(Lines311-312).

Q21. Citation?

Response: Thank you for your suggestions, We have added Citation(Line314).

Q22. Citation?

Response: Thank you for your suggestions, We have added Citation(Line316).

Q23. Please standard the citations in your manuscript, Zhang and Yang, (???)

Response: Thank you for your suggestions, We have improved the Citation in our manuscript(Line328).

Q24. How does the moisture conditions impact the survival of S. italica, please make detailed explanations. 

Response: Thank you for your suggestions, We have made detailed explanations.(Lines332-334).

Q25. Please rewrite the sentence. 

Response: Thank you for your suggestions, We have re-written the sentence(Lines350-351). 

Once again, we sincerely appreciate your significant comments look forward to hearing from you.

Thank your and best regards.

Yours sincerely,

Yi Huang

---

## [Editor Report · Decision Letter 1]

24 Mar 2024

Potential global distribution of Setaria italica , an important species for dryland agriculture in the context of climate change

PONE-D-23-42450R1

Dear Dr. Huang,

We’re pleased to inform you that your manuscript has been judged scientifically suitable for publication and will be formally accepted for publication once it meets all outstanding technical requirements.

Kind regards,

Jiban Shrestha

Academic Editor

PLOS ONE

Additional Editor Comments (optional):

The authors need to check all information carefully.
---

## [Editor Report · Acceptance letter]

3 Apr 2024

PONE-D-23-42450R1 

PLOS ONE

Dear Dr. Huang, 

I'm pleased to inform you that your manuscript has been deemed suitable for publication in PLOS ONE. Congratulations! Your manuscript is now being handed over to our production team.

Kind regards, 

on behalf of

Dr. Jiban Shrestha 

Academic Editor

PLOS ONE